:Ö: PLOS | ONE

# Assessing antigenic drift and phylogeny of influenza A (H1N1) pdm09 virus in Kenya using HA1 sub-unit of the hemagglutinin gene

Silvanos Opanda[1], Wallace Bulimo[1,2]*, George Gachara[3], Christopher Ekuttan[4], Evans Amukoye[5]

1 Department of Emerging Infectious Diseases (DEID), United States Army Medical Research Directorate–Africa (USAMRD-A), Nairobi, Kenya, 2 Department of Biochemistry, University of Nairobi (UoN), Nairobi, Kenya, 3 Department of Medical Laboratory Science, Kenyatta University (KU), Nairobi, Kenya, 4 Defense Forces Memorial Hospital, Kenya Defense Forces (KDF), Nairobi, Kenya, 5 Center for Respiratory Diseases Research, Kenya Medical Research Institute (KEMRI), Nairobi, Kenya

* Wallace.Bulimo@usamru-k.org

**Data Availability Statement:** The sequences of A (H1N1) pdm09 isolates reported in this work are available in GenBank (www.ncbi.nlm.nih.gov/genbank) under accession numbers: ANH22064 -

## Abstract

Influenza A (H1N1) pdm09 virus emerged in North America in 2009 and has been established as a seasonal strain in humans. After an antigenic stasis of about six years, new antigenically distinct variants of the virus emerged globally in 2016 necessitating a change in the vaccine formulation for the first time in 2017. Herein, we analyzed thirty-eight HA sequences of influenza A (H1N1) pdm09 strains isolated in Kenya during 2015–2018 seasons, to evaluate their antigenic and molecular properties based on the HA1 sub-unit. Our analyses revealed that the A (H1N1) pdm09 strains that circulated in Kenya during this period belonged to genetic clade 6B, subclade 6B.1 and 6B.2. The Kenyan 2015 and 2016 isolates differed from the vaccine strain A/California/07/2009 at nine and fourteen antigenic sites in the HA1 respectively. Further, those isolated in 2017 and 2018 correspondingly varied from A/Michigan/45/2015 vaccine strain at three and fifteen antigenic sites. The predicted vaccine efficacy of A/California/07/2009 against Kenyan 2015/2016 was estimated to be 32.4% while A/Michigan/45/2015 showed estimated vaccine efficacies of 39.6% - 41.8% and 32.4% - 42.1% against Kenyan 2017 and 2018 strains, respectively. Hemagglutination-inhibition (HAI) assay using ferret post-infection reference antiserum showed that the titers for the Kenyan 2015/2016 isolates were 2–8-fold lower compared to the vaccine strain. Overall, our results suggest the A (H1N1) pdm09 viruses that circulated in Kenya during 2015/2016 influenza seasons were antigenic variants of the recommended vaccine strains, denoting sub-optimal vaccine efficacy. Additionally, data generated point to a swiftly evolving influenza A (H1N1) pdm09 virus in recent post pandemic era, underscoring the need for sustained surveillance coupled with molecular and antigenic analyses, to inform appropriate and timely influenza vaccine update.

## Introduction

The influenza A (H1N1) pdm09 virus emerged in North America in March 2009 and rapidly spread worldwide causing the first influenza pandemic of the 21st Century [1–4]. The virus

ARK18942; MH316121, MG815810; MH356637 - MH356647 and MK692755 - MK692776. Additional accession numbers in GISAID and GenBank databases of HA gene sequences of A/H1N1 pdm09 reference strains included in the analysis are in the S1 Table.

**Funding:** This work was supported by the US Department of Defense through the Global Emerging Infections Surveillance and Response System (DoD GEIS) for funding the project through Professor Wallace Bulimo, Promis ID 17_KY_1.1.9. The funder had no role in study design, data collection and analysis, decision to publish, or preparation of the manuscript.

**Competing interests:** The authors have declared that no competing interests exist.

genome comprised a rare mixture of gene segments from swine, avian and human influenza viruses [5–8]. The A (H1N1) pdm09 virus was first detected in Kenya on June 2009 [9, 10]. Presently, A (H1N1) pdm09 has become endemic in humans, co-circulating with A/H3N2 and B as seasonal influenza viruses [1, 2, 5, 11]. Influenza virus hemagglutinin (HA) protein is the prime target of host's neutralizing antibodies [12–15]. It is typically cleaved by host cells proteases into HA1 and HA2 domains [16]. Proteolytic cleavage of influenza virus HA protein is crucial for virus infectivity and spread [17]. The A/H1N1 virus HA1 sub-unit comprise five major antigenic sites (epitopes) designated Sa, Sb, Ca1, Ca2 and Cb [18, 19]. However, mapping of H3 epitopes to H1 hemagglutinin (H3 numbering) has provided an alternative approach for identifying antigenic sites in A /H1 viruses [20]. The principal strategy for prevention of influenza disease is vaccination [21, 22]. However, build-up of amino acid mutations coupled with N-linked glycosylation at epitope sites can diminish antibody recognition leading to waning vaccine efficacy and intermittent seasonal epidemics [5, 14, 20, 22, 23]. Since 2010 the World Health Organization (WHO) recommended using A/California/7/2009 as the vaccine component for A (H1N1) pdm09 virus in Northern and Southern hemispheres [2, 21]. This was later replaced in 2017 with A/Michigan/45/2015-like virus due to emergence of antigenic drift variants [2, 15, 24].

The annually formulated influenza vaccine comprises HA antigens from A (H3N2), A (H1N1) pdm09 and one lineage of B (trivalent vaccine) or both lineages (quadrivalent vaccine) as predicted from circulating strains in the preceding seasons [16, 25]. The effectiveness of influenza vaccines fluctuate yearly in part due to genetic evolution of the virus leading to antigenic distance between the vaccine strain and circulating viruses [2, 16, 20, 21]. Different methods including hemagglutination inhibition (HAI) assay and $P_{epitope}$ model may be used to estimate antigenic distance amongst influenza viruses and thus evaluate vaccine efficacy [26, 27]. Nevertheless, previous works have demonstrated that data generated by the $P_{epitope}$ model correlate fairly well with vaccine efficacy [20, 27]. The $P_{epitope}$ model calculates genetic changes in dominant epitopes of the vaccine strain and circulating viruses, providing a measure that corresponds linearly with vaccine efficacy [2, 16, 20, 27]. After an antigenic stasis of about six years characterized by little change in the antigenic epitopes of the HA1 protein, new antigenically distinct variants of the virus emerged in 2016 globally necessitating a change in the vaccine formulation for the first time in the 2017 influenza season [2, 15, 24]. The present study sought to evaluate antigenic and phylogenetic aspects of influenza A (H1N1) pdm09 viruses circulating in Kenya during 2015/2018 when there was increased evolution of the virus, focusing on the HA1 domain of hemagglutinin protein.

## Methods

### Sample collection and preparation

Respiratory nasal swab specimens used in this study were obtained from patients presenting with influenza-like illness (ILI) or severe acute respiratory infection (SARI) symptoms based on the WHO case definitions [28]. They were collected from hospitals comprising human respiratory virus sentinel surveillance network in the Department of Emerging Infectious Diseases (DEID) of the United States Army Medical Research Directorate-Africa (USAMRD-A), within the Kenya Medical Research Institute (KEMRI). The hospital sites are well distributed across the country. The samples were processed at the laboratory as previously described [29].

### Ethics

Two ethical review boards, the Walter Reed Army Institute of Research (WRAIR) Institutional Review Board (IRB) and the Kenya Medical Research Institute (KEMRI) Scientific and Ethics

Review Unit (SERU) reviewed and approved the study protocol under approval numbers WRAIR#1267 and SSC#981, respectively. The study conforms to recognized standards of the US Federal Policy for the Protection of Human Subjects. All participants involved in this study gave consent and consenting of patients prior to sample collection was carried out as previously described [30].

### RNA extraction, RT-PCR and Sanger sequencing

RNA extraction from the isolates was carried out using QIAmp Viral RNA Mini Kit (Qiagen, Inc., USA) according to the manufacturer's instructions. RT-PCR amplification of the HA gene was performed using Superscript III One Step RT-PCR System (Invitrogen Corporation, USA) and a set of previously described M13 tagged primers [31]. A final reaction mix of 25μL containing 2 X Reaction Mix, 20pmoles/μL of both Forward and Reverse primers, Superscript RT/Platinum *Taq* enzyme mix, Nuclease-free water (Promega Corporation, USA) and 3μL RNA template was prepared. The reaction mix was run at 50˚C for 30 min, 94˚C for 2 min followed by 35 cycles of (94˚C for 30 sec, 55˚C for 30 sec, 68˚C for 1 min) and 68˚C for 7 min on a 9700 FAST ABI Thermal Cycler (Applied Biosystems, USA). The PCR amplicons (frag 1: Σ 976 bp; frag 2: Σ 890 bp) were resolved on a 1% Agarose gel (Sigma-Aldrich Co., USA) stained with ethidium bromide (0.5 mg/ml) (Sigma-Aldrich Co., USA) and visualized using the E-box gel documentation system (Vilber Lourmat, France) according to the manufacturer's instructions. The PCR products were cleaned using Exonuclease I/Shrimp Alkaline Phosphatase (ExoSap-IT) enzyme (Affymetrix, USA) and sequenced directly on both strands using universal M13 forward and reverse primers, as previously described [29, 31]. Cycle sequencing was carried out using the Big Dye Terminator Cycle sequencing kit v3.1 (Applied Biosystems, USA) and products were resolved on an automated 3500xL Genetic Analyzer (Applied Biosystems, USA) according to the manufacturer's instructions.

### Nucleotide sequence accession numbers

The sequences of A (H1N1) pdm09 isolates reported in this work are available in GenBank (www.ncbi.nlm.nih.gov/genbank) under accession numbers: ANH22064—ARK18942; MH316121, MG815810; MH356637—MH356647 and MK692755—MK692776.

### Phylogenetic analyses

Nucleotide sequence fragments were processed into contigs using DNA baser v3.2 [32] and aligned with Muscle v3.8 software [33]. A phylogenetic tree was constructed using Bayesian Markov Chain Monte Carlo (MCMC) inference method implemented in MrBayes v3.2 software [34], under the best fit HKY+G nucleotide substitution model as predicted by the jModelTest software [35]. The MCMC was run for 10 million generations, with sampling every 1000 generations and a 10% burn-in. Sequences of relevant reference strains comprising those of known clades and WHO recommended vaccine strains for the southern hemisphere [A/California/7/2009 (H1N1) pdm09-like virus (2015–2016) & A/Michigan/45/2015 (H1N1) pdm09-like virus (2017–2018)] were retrieved from GenBank/ GISAID databases (S1 Table) and included in the analyses.

### Natural selection pressure

Adaptive selection pressures within the HA1 domain were inferred from the ratio of non-synonymous to synonymous changes (dN/dS) using methods available in Datamonkey web server [36]. Selection pressure across the HA1 (mean dN/dS = ω) and specific codon sites were

estimated by the single likelihood ancestor counting (SLAC) and fixed effects likelihood (FEL) methods [37]. The dN/dS ratio was calculated based on neighbor-joining trees under the HKY85 substitution model [38]. Strong evidence of selection was accepted at a P-value <0.05.

### Prediction of N-glycosylation sites

Prediction of potential N-glycosylation sites (amino acid series: Asparagine -X-Serine/Threonine, where X stands for any amino acid except Aspartate or Proline) in the HA1 domain of hemagglutinin protein was carried out using the online NetNGlyc 1.0 server [39]. A score cut-off value of > 0.5 was considered suggestive of glycosylation.

### Determination of vaccine efficacy using $P_{epitope}$ model

The vaccine efficacy of A (H1N1) pdm09 was predicted using the $P_{epitope}$ model [20, 27]. $P_{epitope}$ is expressed as a ratio of amino acid changes in the dominant HA epitope between the vaccine strain and circulating virus [26, 27]. It is computed by dividing the number of amino acid changes in the HA epitope by the total number of amino acids in the epitope) [16, 26, 27]. An epitope with the highest substitutions (antigenic distance) is considered dominant [20, 27]. The epitope regions of A (H1N1) pdm09 isolates analyzed in this study were predicted using H3 numbering as previously described [2, 20]. The vaccine efficacy was calculated by E = (0.47–2.47 x $P_{epitope}$) x 100 [2, 20, 27].

### Hemagglutination-inhibition (HAI) assay

To assess the antigenic relatedness between A (H1N1) pdm09 strains isolated in Kenya and vaccine viruses A/California/7/2009 and A/Michigan/45/2015, hemagglutination-inhibition (HAI) assay was performed using the WHO Collaborating Centre for Reference and Research on Influenza (VIDRL, Melbourne, Australia) HAI typing assay kit, according to the manufacturer's instructions (http://www.influenzacentre.org/flucentres_HIassay.htm).

## Results

Thirty-eight (38) HA sequences of A (H1N1) pdm09 strains isolated in Kenya during 2015 (N = 5), 2016 (N = 2), 2017 (N = 2) and 2018 (N = 29) were analyzed. Homology analysis based on HA1 showed that Kenyan 2015/2016 strains shared 97.6–98.4% (nucleotide) and 93.3–97.2% (amino acid) sequence identities with the vaccine strain A/California/7/2009, whereas the 2017/2018 strains had 97.9–99.4% (nucleotide) and 97.5–99% (amino acid) identities with A/Michigan/45/2015 vaccine virus (Table 1).

Phylogenetic analysis based on HA1 domain of hemagglutinin gene showed that all A (H1N1) pdm09 viruses circulating in Kenya during 2015/2018 belonged to genetic clade 6B; subclade 6B.2 (2015/2016 isolates) and 6B.1 (2017/2018 isolates) (Fig 1). Compared to the vaccine strain A/California/7/2009, Kenyan 2015/2016 strains showed several amino acid changes

**Table 1. Sequence homology of HA1 domain of A (H1N1) pdm09 strains isolated in Kenya relative to WHO vaccine strains.**

| Year | No of strains | Vaccine strain | % identity in HA1 domain | |
|------|---------------|----------------|--------------------------|--------------------------|
| | | | Nucleotide | Amino acid |
| 2015 | 5 | A/California/7/2009 | 97.6–98.4 | 93.3–97.2 |
| 2016 | 2 | A/California/7/2009 | 98–98.2 | 96.6 |
| 2017 | 2 | A/Michigan/45/2015 | 99.3–99.4 | 98.7–99 |
| 2018 | 29 | A/Michigan/45/2015 | 97.9–99.4 | 97.5–99 |

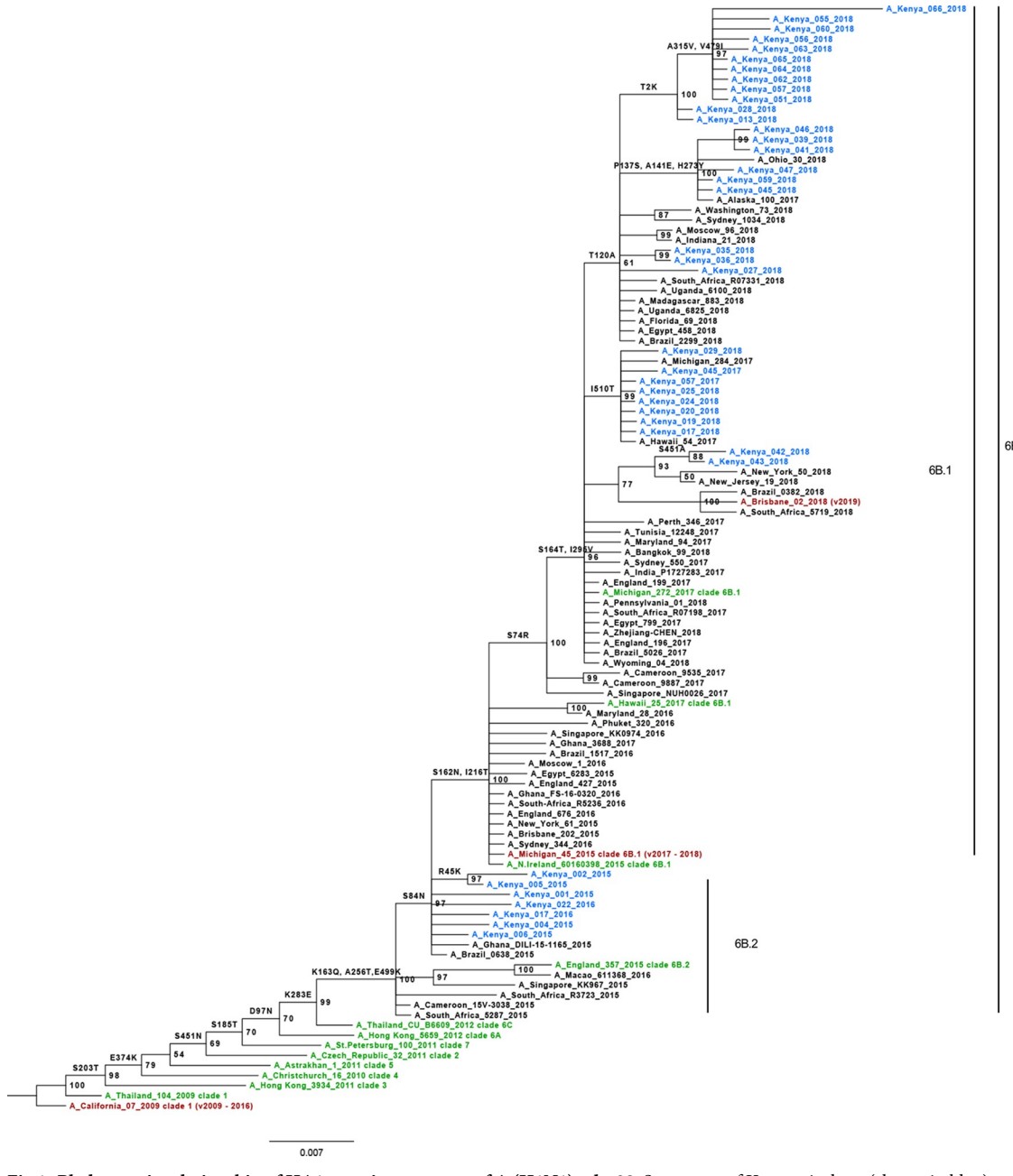

**Fig 1. Phylogenetic relationship of HA1 protein sequences of A (H1N1) pdm09.** Sequences of Kenyan isolates (shown in blue) were analyzed relative to reference strains of known clades (shown in green), vaccine reference strains for southern hemisphere (shown in red) and other reference strains (shown in black). The tree was re-constructed using MrBayes v3.2 with a HKY+G nucleotide substitution model. Numbers at the nodes represent percentage posterior probability values while the scale bar indicates number of amino acid substitutions per site.

in HA1 including S84N (antigenic site E), S203T, E374K, S451N, S185T (antigenic site B), D97N, K283E (antigenic site C), S162R (antigenic site B), K163Q (antigenic site D), A256T and E499K (Fig 1). Similarly, relative to A/Michigan/45/2015 vaccine virus, Kenyan 2017 strains showed S74R, S162N (antigenic site E), S164T (antigenic site D), I295V (antigenic site C) and I510T amino acid variations in HA1, whereas the 2018 strains displayed additional

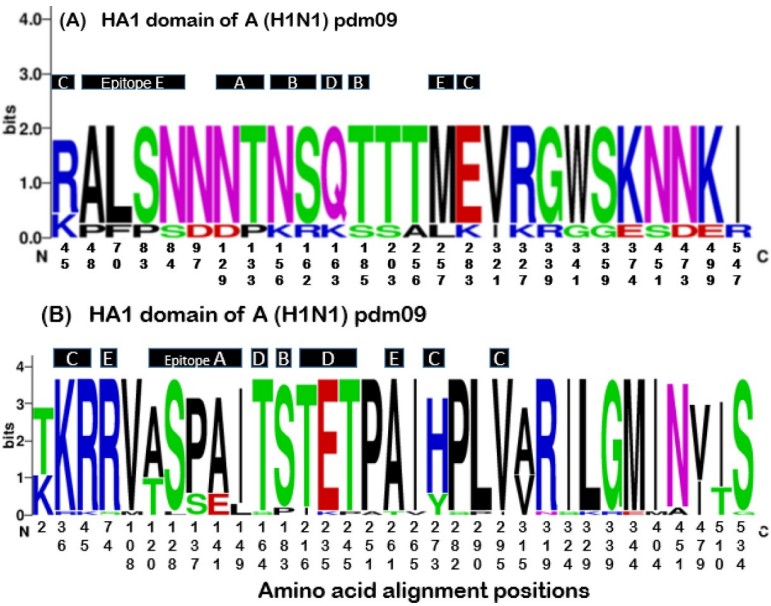

**Fig 2. WebLogo depicting frequency of amino acid changes at the epitope sites (A-E) within the HA1 protein of Kenyan influenza A (H1N1) pdm09 strains isolated between 2015 and 2018.** Amino acid alignment positions along the x-axis in (A) indicate variable sites among Kenyan 2015–2016 strains relative to the vaccine strain A/California/ 2009 while those in (B) depict variable sites among the 2017–2018 isolates relative to the vaccine strain A/Michigan/45/ 2015. The height of the residue indicates the relative frequency of each amino acid at that particular position. These graphics were created using WebLogo (https://weblogo.berkeley.edu/).

T120A (antigenic site A), S451A (n = 2), K36R (n = 1), P137S (n = 6) (antigenic site A), A141E (n = 6) (antigenic site A), H273Y (n = 6) (antigenic site C), T2K (n = 12), I149L (n = 3) (antigenic site A), A315V and V479I amino acid mutations (Figs 1 and 2). Noteworthy, the bulk of these amino acid changes were not unique to Kenyan isolates as they were also observed among the sequences of foreign strains included in the study (S1–S4 Figs).

Natural selection analysis by the SLAC method showed that the mean dN/dS ($\omega$) value for Kenyan A (H1N1) pdm09 strains in the HA1 sub-unit was 0.65 (95% CI), suggesting negative (purifying) selection. No specific codon sites within the HA1 sub-unit were detected to be evolving under negative or positive selection by either the SLAC or FEL methods at P-value < 0.05. All 2015/2016 Kenyan strains retained four potential N-glycosylation sites (at amino acid residue positions 11, 23, 87 and 287) present in the vaccine strain A/California/7/2009. The 2017/2018 strains (with exception of a strain designated A/Kenya/066/2018, accession no: MK692755) had an extra N-glycosylation site (at amino acid residue position 162) present in vaccine component A/Michigan/45/2015. The gain of a glycosylation motif in 2017/2018 A (H1N1) pdm09 viruses was due to a S162N amino acid substitution.

The predicted vaccine efficacies of vaccine strains A/California/7/2009 and A/Michigan/ 45/2015 against Kenyan 2015/2016 and 2017/2018 strains are summarized in Table 2. The P$_{epitope}$ between vaccine strain A/California/7/2009 and Kenyan 2015/2016 strains was 0.059 (dominant epitope E; substitutions P83S and S84N), suggesting a predicted vaccine efficacy of 32.4% of that of a perfect match with the vaccine strain. The P$_{epitope}$ between the vaccine strain A/Michigan/45/2015 and Kenyan 2017 strains ranged from 0.021–0.03, suggesting a predicted vaccine efficacy of 39.6–41.8%. Antigenic drift in Kenyan 2018 strains was observed mainly on epitopes C, D and E, yielding a predicted vaccine efficacy range of 32.4–42.1%. The predicted vaccine efficacy of the 2019 influenza A H1N1 pdm09 vaccine strain component (A/Brisbane/

**Table 2. Estimated vaccine efficacy of recommended WHO vaccine strains against influenza A (H1N1) pdm09 strains circulating in Kenya during 2015 to 2018 seasons.** Dominant epitopes are shown in Bold.

| Year (N) | Vaccine Strain | No. of strains | Dominant Epitope | No. of mutations | Residue differences | Pepitope | Efficacy (%) |
|---|---|---|---|---|---|---|---|
| 2015 (N = 5) | A/California/7/2009 | 2 | C | 1 | R45K | 0.03 | 39.6 |
| | | 5 | C | 1 | K283E | 0.03 | 39.6 |
| | | 1 | E | 1 | A48P | 0.029 | 39.8 |
| | | **5** | **E** | **2** | **P83S, S84N** | **0.059** | **32.4** |
| | | 1 | A | 2 | N129D | 0.042 | 36.6 |
| | | 1 | A | 1 | T133P | 0.042 | 36.6 |
| | | 5 | D | 1 | K163Q | 0.021 | 41.8 |
| | | 5 | B | 1 | S185T | 0.045 | 35.9 |
| 2016 (N = 2) | A/California/7/2009 | 1 | E | 1 | L70F | 0.029 | 39.8 |
| | | **2** | **E** | **2** | **P83S, S84N** | **0.059** | **32.4** |
| | | 1 | E | 1 | M257L | 0.029 | 39.8 |
| | | 1 | B | 1 | N156K | 0.045 | 35.9 |
| | | 1 | B | 1 | S162R | 0.045 | 35.9 |
| | | 2 | B | 1 | S185T | 0.045 | 35.9 |
| | | 2 | D | 1 | K163Q | 0.021 | 41.8 |
| | | 2 | C | 1 | K283E | 0.03 | 39.6 |
| 2017 (N = 2) | A/Michigan/45/2015 | **2** | **E** | **1** | **S74R** | **0.029** | **39.8** |
| | | **2** | **D** | **1** | **S164T** | **0.021** | **41.8** |
| | | **2** | **C** | **1** | **I295V** | **0.03** | **39.6** |
| 2018 (N = 29) | A/Michigan/45/2015 | 1 | C | 1 | K36R | 0.03 | 39.6 |
| | | 1 | C | 1 | R45K | 0.03 | 39.6 |
| | | 6 | C | 1 | H273Y | 0.03 | 39.6 |
| | | **29** | **C** | **1** | **I295V** | **0.03** | 39.6 |
| | | **29** | **E** | **1** | **S74R** | **0.059** | 32.4 |
| | | 1 | E | 1 | A261T | 0.029 | 39.8 |
| | | 2 | B | 1 | S183P | 0.045 | 35.9 |
| | | 21 | A | 1 | T120A | 0.042 | 36.6 |
| | | 1 | A | 1 | S128L | 0.042 | 36.6 |
| | | 6 | A | 2 | P137S, A141E | 0.083 | 26.5 |
| | | 3 | A | 1 | I149L | 0.042 | 36.6 |
| | | **29** | **D** | **1** | **S164T** | **0.02** | **42.1** |
| | | 2 | D | 1 | T216I | 0.02 | 42.1 |
| | | 1 | D | 1 | E235K | 0.02 | 42.1 |
| | | 1 | D | 1 | T245P | 0.02 | 42.1 |
| | A/Brisbane/02/2018 | 21 | A | 1 | T120A | 0.042 | 36.6 |
| | | 6 | A | 2 | P137S, A141E | 0.083 | 26.5 |
| | | 1 | A | 1 | I149L | 0.042 | 36.6 |
| | | 1 | A | 1 | S128L | 0.042 | 36.6 |
| | | **29** | **C** | **2** | **G45R, V298I** | **0.06** | **32.2** |
| | | 6 | C | 1 | H273Y | 0.03 | 39.6 |
| | | 1 | C | 1 | K36R | 0.03 | 39.6 |
| | | **29** | **D** | **1** | **R223Q** | **0.02** | **42.1** |
| | | 1 | D | 1 | T245P | 0.02 | 42.1 |
| | | 2 | D | 1 | T216I | 0.02 | 42.1 |
| | | 27 | B | 1 | P183S | 0.045 | 35.9 |
| | | 1 | E | 1 | A261T | 0.059 | 32.4 |

02/2018) against Kenyan 2018 strains yielded vaccine efficacy range of 32.2–42.1% of that of a perfect match, with mutations observed mostly on epitopes C and D. Remarkably, the vaccine effectiveness between A/Michigan/45/2015 and A/Brisbane/02/2018 vaccine strains against Kenyan 2018 A/H1N1 pdm09 isolates fell within the same margin (Σ 32–42%), plausibly implying that the latter A/H1N1 pdm09 vaccine component would not have provided improved protection against these viruses.

A summary of the HAI assay titers for Kenyan A/H1N1 pdm09 isolates and vaccine viruses A/California/7/2009 (2015/2016 influenza seasons) and A/Michigan/45/2015 (2017/2018 influenza seasons) is shown in Table 3. The Kenyan 2015/2016 isolates exhibited lower HAI titers with the A/California/7/2009 reference antiserum relative to the vaccine antigen A/California/7/2009, suggesting reduced antibody recognition and binding to antigenic sites among the Kenyan viruses. Conversely, all the Kenyan isolates, including those obtained in 2017/2018 exhibited higher HAI assay titers with the A/Michigan/45/2015 reference antiserum, suggesting enhanced recognition and binding to antigenic sites among the study isolates.

## Discussion

In line with global circulation trend [2, 11, 21, 24, 40–44], we have shown that influenza A (H1N1) pdm09 viruses that circulated in Kenya during 2015–2018 belonged to phylogenetic clade 6B; sub-clades 6B.1 and 6B.2. Clade 6B of the A (H1N1) pdm09 variants are defined by D97N, K163Q (antigenic site D), S185T (antigenic site B), K283E (antigenic site C) and A256T amino acid mutations in HA1 while those belonging to sub-clade 6B.1 harbor additional S74R (antigenic site E), S162N (antigenic site B), S164T (antigenic site D), I216T (antigenic site D) and I295V (antigenic site C) amino acid substitutions [11, 20, 21, 42]. Influenza A (H1N1) pdm09 clade 6B/6B.1/6B.2 variants have been associated with increased disease severity compared to non-6B/6B.1/6B.2 members, partly due to antigenic alterations the former have undergone [11].

Scrutiny of our analyzed data revealed nine to thirteen antigenic amino acid alterations in HA1 sub-unit of Kenyan 2015/2016 strains compared to the vaccine strain A/California/7/2009, parallel to 2018 strains which showed fifteen disparate antigenic variations with respect to A/Michigan/45/2015 vaccine virus. This result indicates that influenza A (H1N1) pdm09 viruses that circulated in Kenya during 2015/2016 and 2018 seasons were genetic variants of the vaccine strains and reverberates findings of similar studies reported elsewhere [11, 21, 45]. Previous works have shown that four or more amino acid modifications at different HA1 epitope sites of an influenza virus is antigenically significant as it can give rise to new drift variants with altered antigenic properties [29, 46, 47]. These observations suggest that besides being genetic variants of the vaccine strains, the Kenyan viruses were antigenically divergent from those vaccine strains, plausibly implying that the vaccines provided sub-optimal protection against the A (H1N1) pdm09 viruses.

Further analyses showed that consistent with previous findings [2, 42, 48] all potential N-linked glycosylation sites were conserved in all but one influenza A (H1N1) pdm09 strains circulating in Kenya during 2015/2018 seasons, comparative to respective vaccine strains. All analyzed Kenyan A (H1N1) pdm09 subclade 6B.1 members displayed an additional N-glycosylation at residue sites 162–164 attributed to the S162N substitution [42]. The potential gain or loss of N-linked glycosylation at HA1 epitope sites can alter influenza virus antigenic properties [13, 49].

The average dN/dS (ω) value indicated that the A (H1N1) pdm09 viruses circulating in Kenya during 2015/2018 were evolving under purifying selection in HA1 protein. Additionally, no specific codon sites in this hemagglutinin protein region were found to be positively or negatively selected. Our result is consistent to findings by Tewawong et al., (2015) [16].

**Table 3. HAI assay titers of A (H1N1) pdm09 strains isolated in Kenya and vaccine antigens with WHO reference antisera for 2015/2016 and 2017/2018 influenza seasons.** The numbers indicate HAI titers.

| Reference Antigens | Reference Antisera | |
|---|---|---|
| | A/California/7/2009 (2015/2016) | A/Michigan/45/2015 (2017/2018) |
| A/California/7/2009 (2015/2016) | 256 | 512 |
| A/Michigan/45/2015 (2017/2018) | 128 | 1024 |
| Test isolates | | |
| A/Kenya/001/2015 | 64 | 512 |
| A/Kenya/002/2015 | 128 | 1024 |
| A/Kenya/004/2015 | 64 | 512 |
| A/Kenya/005/2015 | 64 | 512 |
| A/Kenya/006/2015 | 32 | 256 |
| A/Kenya/013/2018 | 128 | 1024 |
| A/Kenya/017/2016 | 32 | 256 |
| A/Kenya/017/2018 | 256 | 1024 |
| A/Kenya/019/2018 | 64 | 512 |
| A/Kenya/020/2018 | 256 | 1024 |
| A/Kenya/022/2016 | 64 | 512 |
| A/Kenya/024/2018 | 128 | 512 |
| A/Kenya/025/2018 | 128 | 512 |
| A/Kenya/027/2018 | 64 | 512 |
| A/Kenya/028/2018 | 32 | 256 |
| A/Kenya/029/2018 | 64 | 256 |
| A/Kenya/035/2018 | 128 | 512 |
| A/Kenya/036/2018 | 64 | 256 |
| A/Kenya/039/2018 | 64 | 512 |
| A/Kenya/041/2018 | 128 | 512 |
| A/Kenya/042/2018 | 64 | 512 |
| A/Kenya/043/2018 | 256 | 512 |
| A/Kenya/045/2017 | 64 | 512 |
| A/Kenya/045/2018 | 64 | 512 |
| A/Kenya/046/2018 | 64 | 512 |
| A/Kenya/047/2018 | 64 | 512 |
| A/Kenya/051/2018 | 128 | 512 |
| A/Kenya/055/2018 | 64 | 256 |
| A/Kenya/056/2018 | 64 | 512 |
| A/Kenya/057/2017 | 128 | 512 |
| A/Kenya/057/2018 | 32 | 256 |
| A/Kenya/059/2018 | 128 | 512 |
| A/Kenya/060/2018 | 64 | 256 |
| A/Kenya/062/2018 | 64 | 256 |
| A/Kenya/063/2018 | 64 | 256 |
| A/Kenya/064/2018 | 64 | 256 |
| A/Kenya/065/2018 | 64 | 256 |
| A/Kenya/066/2018 | 64 | 512 |

The predicted vaccine efficacy of A/California/7/2009 vaccine strain against influenza A (H1N1) pdm09 strains circulating in Kenya during 2015/2016 season was 32.4%. The sub-optimal vaccine efficacy suggests antigenic drift and corresponds with our genetic analysis data.

This result corroborates findings of previous works and buttresses the WHO September 2016 decision to update southern hemisphere A (H1N1) pdm09 vaccine strain [21, 22, 41, 45]. The estimated vaccine efficacy between A/Michigan/45/2015 vaccine strain and influenza A (H1N1) pdm09 viruses circulating in Kenya during 2017 season ranged from 39.6 to 41.8%. These estimates imply improved vaccine efficacy and reflect findings of related studies conducted elsewhere [50] Subsequently, the estimated vaccine efficacy of A/Michigan/45/2015 vaccine strain against Kenyan 2018 viruses ranged from 32.4% to 42.1%, indicating a decrease in vaccine effectiveness compared to the preceding year. This observation is attributed to several antigenic amino acid alterations noticed among Kenyan 2018 strains and yet again supports the WHO February 2019 decision to update influenza A (H1N1) pdm09 virus vaccine component for northern hemisphere during the 2019–2020 season A/Brisbane /02/2018 [51]. Assessment of the vaccine efficacy of A/Brisbane /02/2018 vaccine strain against Kenyan 2018 viruses indicated a predicted vaccine efficacy range of 32.2% - 42.1%, suggesting that the new vaccinating virus will not provide improved protection against the circulating viruses.

To investigate whether antigenic drift observed among the Kenyan A/H1N1 pdm09 strains was attributable to the lower vaccine efficacy predictions, HAI assay characterization of the isolates using reference post-infection ferret antiserum was conducted. Data generated revealed that the A/H1N1 pdm09 strains isolated in Kenya during the 2015/2016 influenza seasons had lower HAI titers ($\geq$2-fold) compared to the vaccine strain, suggesting that despite reports indicating that the A/H1N1 pdm09 viruses that circulated in the world during this period were antigenically similar to A/California/7/2009 [52, 53], those circulating in Kenya had undergone considerable antigenic divergence from the vaccine strain, leading to sub-optimal vaccine efficacy. This finding is consistent with that of a similar study reported elsewhere [54], and supports the WHO September 2016 decision to update the A/H1N1 pdm09 strain component of influenza vaccine from A/California/7/2009 to A/Michigan/45/2015 [53]. Consequently, those viruses isolated during 2017/2018 exhibited high HAI titers with the reference antiserum, indicating higher antigenic relatedness with the vaccine strain A/Michigan/45/2015 and improved vaccine efficacy.

This study had a few shortcomings. First, the small number of sequences analyzed for viruses that circulated in Kenya during 2015 /2017 was somewhat not representative. Secondly, we predicted antigenic drift and vaccine efficacy using H3 numbering system [20] since there is yet to be a consensus method for $P_{epitope}$ calculation in A (H1N1) pdm09 viruses [2, 16, 27]. Despite these shortcomings, we have provided evidence that influenza A (H1N1) pdm09 viruses isolated in Kenya during 2015/2018 belonged to phylogenetic clade 6B, subclades 6B.2 and 6B.1. Moreover, we have demonstrated that the A (H1N1) pdm09 vaccine strain A/ California/07/2009 exhibited sub-optimal vaccine efficacy against Kenyan 2015/2016 strains, denoting antigenic drift among these viruses. Overall, these findings indicate a swiftly evolving A (H1N1) pdm09 virus in recent post pandemic era, underscoring the need for sustained surveillance coupled with molecular and antigenic analyses, to inform appropriate and timely influenza vaccine update.

## Supporting information

**S1 Table. Accession numbers in GISAID and GenBank databases of HA gene sequences of A/H1N1 pdm09 reference strains included in the analysis.**
(DOCX)

**S1 Fig. Alignment of HA1 amino acid sequences of A/H1N1 pdm09 strains isolated in Kenya in 2015 with foreign strains, relative to vaccine virus A/California/7/2009.**
(PDF)

**S2 Fig. Alignment of HA1 amino acid sequences of A/H1N1 pdm09 strains isolated in Kenya in 2016 with foreign strains relative to vaccine virus A/California/7/2009.** (PDF)

**S3 Fig. Alignment of HA1 amino acid sequences of A/H1N1 pdm09 strains isolated in Kenya in 2017 with foreign strains relative to vaccine virus A/Michigan/45/2015.** (PDF)

**S4 Fig. Alignment of HA1 amino acid sequences of A/H1N1 pdm09 strains isolated in Kenya in 2018 with foreign strains relative to vaccine virus A/Michigan/45/2015.** (PDF)

## Acknowledgments

We are grateful for patients who provided specimens from which the viruses were isolated.

Disclaimer: Material has been reviewed by the Walter Reed Army Institute of Research (WAIR). There is no objection to its presentation and/or publication. The opinions or assertions contained herein are the private views of the authors, and are not to be construed as official, or as reflecting true views of the Department of the Army or the Department of Defense. The investigators have adhered to the policies for protection of human subjects as prescribed in AR 70–25.

## Author Contributions

**Conceptualization:** Silvanos Opanda, Wallace Bulimo, Christopher Ekuttan.

**Data curation:** Silvanos Opanda, George Gachara.

**Formal analysis:** Silvanos Opanda.

**Funding acquisition:** Wallace Bulimo.

**Methodology:** Silvanos Opanda.

**Project administration:** Christopher Ekuttan, Evans Amukoye.

**Supervision:** Wallace Bulimo.

**Writing – original draft:** Silvanos Opanda.

**Writing – review & editing:** Wallace Bulimo, George Gachara, Evans Amukoye.

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
