## [Decision Letter · Decision Letter 0]

12 Nov 2019

PONE-D-19-29545

Assessing Antigenic Drift and Phylogeny of Influenza A (H1N1) pdm09 Virus in Kenya the Using HA1 sub-unit of the Hemagglutinin gene.

PLOS ONE

Dear Mr Opanda,

Thank you for submitting your manuscript to PLOS ONE. After careful consideration, we feel that it has merit but does not fully meet PLOS ONE’s publication criteria as it currently stands. Therefore, we invite you to submit a revised version of the manuscript that addresses the points raised during the review process.

We would appreciate receiving your revised manuscript by Dec 27 2019 11:59PM. To enhance the reproducibility of your results, we recommend that if applicable you deposit your laboratory protocols in protocols.io, where a protocol can be assigned its own identifier (DOI) such that it can be cited independently in the future. For instructions see: http://journals.plos.org/plosone/s/submission-guidelines#loc-laboratory-protocols

We look forward to receiving your revised manuscript.

Kind regards,

Ronald Dijkman, PhD

Academic Editor

PLOS ONE

Journal Requirements:

2. Please amend the manuscript submission data (via Edit Submission) to include authors Wallace Bulimo, George Gachara, Christopher Ekuttan and Evans Amukoye.

Additional Editor Comments:

Dear Author,

Both reviewers and myself, highly recommend you to include HAI antigenic data in the revised version to complement the current in silico data.

Thank you for your time and understanding.

Reviewers' comments:

Reviewer's Responses to Questions

**Comments to the Author**

1. Is the manuscript technically sound, and do the data support the conclusions?

Reviewer #1: Partly

Reviewer #2: Partly

2. Has the statistical analysis been performed appropriately and rigorously? 

Reviewer #1: N/A

Reviewer #2: Yes

3. Have the authors made all data underlying the findings in their manuscript fully available?

Reviewer #1: Yes

Reviewer #2: Yes

4. Is the manuscript presented in an intelligible fashion and written in standard English?

Reviewer #1: Yes

Reviewer #2: Yes

5. Review Comments to the Author

Reviewer #1: In their manuscript, Opanda et. al. describe an in silico study of A/H1N1 viruses circulating in Kenya from 2015-2018. The paper is well written, clear, and the data is important in showing the natural drift in influenza A/H1N1 viruses in Kenya and issues that might arrive from decreased vaccine effectiveness. However, to increase the novelty of the manuscript, more work should be done with data available from other sources as well as in vitro analysis to confirm these in silico results. A few major comments for improvement exist.

1) It is unclear if the strains used in this analysis are new, or are what is already available on the Influenza Research database (fludb.org) and on GISAID (gisaid.org)…if these are new strains, then the 46 available H1N1 strains from Kenya on IRD with addition of non-duplicates from the 121 strains available on GISAID should be included in this study as it will greatly increase the power of the analyses performed. This will alleviate the first limitation of the study, somewhat.

2) Ideally, it would be good to show HAI antigenic data (and even microneutralization data using reference serum) to show how these in silico results correlate to wet laboratory results. Indeed, antigenic cartography of these viruses would be a good way to further show drift and possible vaccine escape. This will definitely alleviate the second shortcoming of the study as mentioned in the discussion.

3) Lines 241-244: This is extremely important information and should be highlighted more in the results, as it is not discussed previously.

4) While calculating the predicted efficacy of the vaccine is useful (under ideal conditions, of course) ,it would be very good to compare the results from this study with the vaccine effectiveness calculated in Kenya for each year so as to show that these predictive values d hold true to their intended results.

5) How does this predicted drift compare to viruses isolated in the greater geographic area? Is there some hypothesis as to why the viruses in Kenya appear to be drifting in such a manner?

Reviewer #2: Overall, the manuscript is well written the main points are clearly stated. The use of bibliography is adequate and the tables and figures are sufficiently clear to help interpret the results.

Comments:

-line 45: The first pandemic of the 21st century was SARS, please amend.

-line 102: what primers? How large is the resulting fragment? Does it cover the complete ORF?

-line 111: M13 sequencing primers: where is the description of the cloning vector and process used to sequence with the M13 primers?

-line 123: What inference method was used? Also, how many generations where run and sampled every how many times. Any % burn-ins excluded? It is also important, going back to line 102, how large the gene fragments are to asses how reliable the analysis is.

-line 125: please provide accession numbers in supplemental material

-line 142: I would indicate that you should also consider the NGlyc result in your estimation of glycosylated sites. The algorithm is very specific in that a >/= score of (++) should be used for describing an N-glycosylation site with high specificity. I would include that criteria as well and only inform those sites that also meet that criteria. A bit more conservative, since you are using only computational analyses throughout this manuscript.

-line 247: It is a good practice to state the shortcomings of a manuscript, however, you should also give a reason why. In this case it would have been very interesting to see how the HAI results correlate to the results of this manuscript. HAI is a relatively simple procedure if you have the reagents. At least give an explanation of why it was not incorporated.

6. PLOS authors have the option to publish the peer review history of their article (what does this mean?). If published, this will include your full peer review and any attached files.

Reviewer #1: No

Reviewer #2: No

---

## [Author Response · Author response to Decision Letter 0]

27 Dec 2019

Below is a point to point response to the comments by the academic editor reviewers:

ACADEMIC EDITOR PLOS ONE:

Please ensure that your manuscript meets PLOS ONE's style requirements, including those for file naming

2. Please amend the manuscript submission data (via Edit Submission) to include authors Wallace Bulimo, George Gachara, Christopher Ekuttan and Evans Amukoye.

3. Both reviewers and myself, highly recommend you to include HAI antigenic data in the revised version to complement the current in silico data.

Thank you, all these have been addressed.

REVIEWER # 1: 

In their manuscript, Opanda et. al. describe an in silico study of A/H1N1 viruses circulating in Kenya from 2015-2018. The paper is well written, clear, and the data is important in showing the natural drift in influenza A/H1N1 viruses in Kenya and issues that might arrive from decreased vaccine effectiveness. However, to increase the novelty of the manuscript, more work should be done with data available from other sources as well as in vitro analysis to confirm these in silico results. A few major comments for improvement exist.

1. It is unclear if the strains used in this analysis are new, or are what is already available on the Influenza Research database (fludb.org) and on GISAID (gisaid.org)…if these are new strains, then the 46 available H1N1 strains from Kenya on IRD with addition of non-duplicates from the 121 strains available on GISAID should be included in this study as it will greatly increase the power of the analyses performed. This will alleviate the first limitation of the study, somewhat.

Thank you for this observation. The Kenyan A/H1N1 pdm09 HA gene sequence data used in this study is not new but are the ones already available in the genetic databases under accession numbers: ANH22064 - ARK18942; MH316121, MG815810; MH356637 - MH356647 and MK692755 - MK692776 in the manuscript (lines 119 – 120). The study focused on the sequence data generated during 2015 to 2018 influenza seasons in Kenya.

2. Ideally, it would be good to show HAI antigenic data (and even microneutralization data using reference serum) to show how these in silico results correlate to wet laboratory results. Indeed, antigenic cartography of these viruses would be a good way to further show drift and possible vaccine escape. This will definitely alleviate the second shortcoming of the study as mentioned in the discussion. 

Thank you for this suggestion. The HAI assay antigenic data has been incorporated in the revised manuscript to validate the in-silico analyses to alleviate the second limitation of the study (Table 3, lines 250 – 253).

3. Lines 241-244: This is extremely important information and should be highlighted more in the results, as it is not discussed previously.

We concur with the reviewer. A statement highlighting the difference in vaccine efficacies between A/Michigan/45/2015 and A/Brisbane/02/2018 vaccine strains against Kenyan 2018 A/H1N1 pdm09 isolates has been incorporated in the revised manuscript results section (lines 231 – 235). Unfortunately, we could to provide HAI assay antigenic data between the Kenyan 2018 isolates and A/Brisbane/02/2018 reference anti-serum, since our lab ceased being recognized as an NIC in 2018 and therefore we no longer receive HAI assay Kits from WHO CC reference labs. 

4. While calculating the predicted efficacy of the vaccine is useful (under ideal conditions, of course), it would be very good to compare the results from this study with the vaccine effectiveness calculated in Kenya for each year so as to show that these predictive values hold true to their intended results. 

Thank you for your suggestion. The calculated vaccine effectiveness across the study period (2015-2018) have been presented in the manuscript in lines 221-235 and summarized in Table 2. However, we would like to point out that no vaccine effectiveness studies involving clinical trials are ever done in Kenya due to cost considerations. Hence, we are unable to compare the calculated vaccine effectiveness against predicted/observed (via clinical trials) vaccine effectiveness for Kenya.

5. How does this predicted drift compare to viruses isolated in the greater geographic area? Is there some hypothesis as to why the viruses in Kenya appear to be drifting in such a manner?

 Actually, the drifting the A/H1N1 pdm09 strains reported here is not unique to Kenyan viruses. The bulk of amino acid changes observed affecting antigenic sites of the HA1 domain among the Kenyan isolates have also been noticed in the sequences of foreign strains included in the study. To address this question, additional supporting information S1 Fig, S2 Fig, S3 Fig and S4 Fig has been added in the revised manuscript. 

REVIEWER # 2: 

1. - Line 45: The first pandemic of the 21st century was SARS, please amend.

Thank you for this observation. This has been amended in the revised manuscript (line 45). 

2. - Line 102: what primers? How large is the resulting fragment? Does it cover the complete ORF?

These are A/H1N1 pdm09 HA segment-specific primers tagged with flanking universal M13 primers to facilitate ease of sequencing. We have included a citation in the revised manuscript to address this (line101). The HA segment was amplified using two sets of primers to yield two overlapping PCR fragments: frag 1: ⁓ 976 bp and frag 2: ⁓ 890 bp. The contig of the two fragments (1.8 kb) covers the complete open reading frame of the HA gene (lines 106 – 107). 

3. -line 111: M13 sequencing primers: where is the description of the cloning vector and process used to sequence with the M13 primers? 

We did not clone the amplicons into a vector. Instead the M13 primer tags were incorporated to provide anchors flanking the amplicons to facilitate sequencing of the amplicons. The statement has been edited accordingly in the revised version of the manuscript providing the source detailing the process of sequencing the PCR amplicons with the M13 primers (lines 111-112).

4. -line 123: What inference method was used? Also, how many generations were run and sampled every how many times. Any % burn-ins excluded? It is also important, going back to line 102, how large the gene fragments are to assess how reliable the analysis is. 

Thank you for this comment. The phylogenetic tree was constructed using Bayesian Monte Carlo Markov Chain (MCMC) inference method. The MCMC was run for 10 million generations, with sampling every 1000 generations and a 10% burn-in. These statements have been incorporated in the revised manuscript (lines 124 – 128). 

5. -line 125: please provide accession numbers in supplemental material. 

Thank you for your suggestion. Accession numbers of HA sequences of all reference strains included in the analysis has been provided as supplemental material (S1 Table). 

6. -line 142: I would indicate that you should also consider the NGlyc result in your estimation of glycosylated sites. The algorithm is very specific in that a >/= score of (++) should be used for describing an N-glycosylation site with high specificity. I would include that criteria as well and only inform those sites that also meet that criteria. A bit more conservative, since you are using only computational analyses throughout this manuscript.

We concur with the reviewer’s comment. A score of (++) describes an N-glycosylation with high specificity. However, as stated in line 146 of the manuscript we considered as score cut-off value of > 0.5 suggestive of an N-glycosylation. We took (+) and (++) scores as evidence of potential glycosylation.

7. -line 247: It is a good practice to state the shortcomings of a manuscript, however, you should also give a reason why. In this case it would have been very interesting to see how the HAI results correlate to the results of this manuscript. HAI is a relatively simple procedure if you have the reagents. At least give an explanation of why it was not incorporated.

Again, we appreciate the reviewer’s comment. We have not only mentioned the shortcomings but also addressed the strengths of the study. The HAI assay antigenic data has been incorporated in the revised manuscript (lines 251 – 254) and they corroborate the PEpitope findings which is reassuring about the power of PEpitope model.

---

## [Decision Letter · Decision Letter 1]

7 Jan 2020

Assessing Antigenic Drift and Phylogeny of Influenza A (H1N1) pdm09 Virus in Kenya Using HA1 sub-unit of the Hemagglutinin gene.

PONE-D-19-29545R1

Dear Dr. Opanda,

All the best wishes for 2020.

We are pleased to inform you that your manuscript has been judged scientifically suitable for publication and will be formally accepted for publication once it complies with all outstanding technical requirements.

With kind regards,

Ronald Dijkman, PhD

Academic Editor

PLOS ONE

Additional Editor Comments (optional):

Reviewers' comments:

Reviewer's Responses to Questions

**Comments to the Author**

1. If the authors have adequately addressed your comments raised in a previous round of review and you feel that this manuscript is now acceptable for publication, you may indicate that here to bypass the “Comments to the Author” section, enter your conflict of interest statement in the “Confidential to Editor” section, and submit your "Accept" recommendation.

Reviewer #1: All comments have been addressed

Reviewer #2: All comments have been addressed

2. Is the manuscript technically sound, and do the data support the conclusions?

Reviewer #1: Yes

Reviewer #2: Yes

3. Has the statistical analysis been performed appropriately and rigorously? 

Reviewer #1: Yes

Reviewer #2: Yes

4. Have the authors made all data underlying the findings in their manuscript fully available?

Reviewer #1: Yes

Reviewer #2: Yes

5. Is the manuscript presented in an intelligible fashion and written in standard English?

Reviewer #1: Yes

Reviewer #2: Yes

6. Review Comments to the Author

Reviewer #1: (No Response)

Reviewer #2: (No Response)

7. PLOS authors have the option to publish the peer review history of their article (what does this mean?). If published, this will include your full peer review and any attached files.

Reviewer #1: No

Reviewer #2: No

---

## [Editor Report · Acceptance letter]

3 Feb 2020

PONE-D-19-29545R1 

Assessing Antigenic Drift and Phylogeny of Influenza A (H1N1) pdm09 Virus in Kenya Using HA1 sub-unit of the Hemagglutinin gene. 

Dear Dr. Opanda:

I am pleased to inform you that your manuscript has been deemed suitable for publication in PLOS ONE. Congratulations! Your manuscript is now with our production department. 

With kind regards,

on behalf of

Dr. Ronald Dijkman 

Academic Editor

PLOS ONE